# The Role of Continuous Glucose Monitoring, Diabetes Smartphone Applications, and Self-Care Behavior in Glycemic Control: Results of a Multi-National Online Survey

**DOI:** 10.3390/jcm8010109

**Published:** 2019-01-17

**Authors:** Mihiretu M. Kebede, Cora Schuett, Claudia R. Pischke

**Affiliations:** 1Health Sciences, University of Bremen, Grazerstrasse 2, D-28359 Bremen, Germany; 2Leibniz Institute for Prevention Research and Epidemiology—BIPS, Achterstrasse 30, D-28359 Bremen, Germany; coraschuett@gmail.com (C.S.); ClaudiaRuthPischke@med.uni-duesseldorf.de or claudia.pischke@leibniz-bips.de (C.R.P.); 3Institute of Public Health, College of Medicine and Health Science, University of Gondar, Po.box-196 Gondar, Ethiopia; 4Institute of Medical Sociology, Centre for Health and Society, Medical Faculty, University of Duesseldorf, Universitätsstrasse 1, D-40225 Duesseldorf, Germany

**Keywords:** diabetes, CGM, self-care, glycemic control, hyperglycemia, hypoglycemia

## Abstract

**Background:** This study investigated the determinants (with a special emphasis on the role of diabetes app use, use of continuous glucose monitoring (CGM) device, and self-care behavior) of glycemic control of type 1 and type 2 diabetes mellitus (DM). **Methods:** A web-based survey was conducted using diabetes Facebook groups, online patient-forums, and targeted Facebook advertisements (ads). Demographic, CGM, diabetes app use, and self-care behavior data were collected. Glycemic level data were categorized into hyperglycemia, hypoglycemia, and good control. Multinomial logistic regression stratified by diabetes type was performed. **Results:** The survey URL was posted in 78 Facebook groups and eight online forums, and ten targeted Facebook ads were conducted yielding 1854 responses. Of those owning smartphones (*n* = 1753, 95%), 1052 (62.6%) had type 1 and 630 (37.4%) had type 2 DM. More than half of the type 1 respondents (*n* = 549, 52.2%) and one third the respondents with type 2 DM (*n* = 210, 33.3%) reported using diabetes apps. Increased odds of experiencing hyperglycemia were noted in persons with type 1 DM with lower educational status (Adjusted Odds Ratio (AOR) = 1.7; 95% Confidence Interval (CI): 1.21–2.39); smokers (1.63, 95% CI: 1.15–2.32), and high diabetes self-management concern (AOR = 2.09, 95% CI: 1.15–2.32). CGM use (AOR = 0.66, 95% CI: 0.44–1.00); “general diet” (AOR = 0.86, 95% CI: 0.79–0.94); and “blood glucose monitoring” (AOR = 0.88, 95%CI: 0.80–0.97) self-care behavior reduced the odds of experiencing hyperglycemia. Hypoglycemia in type 1 DM was reduced by using CGM (AOR = 0.24, 95% CI: 0.09–0.60), while it was increased by experiencing a high diabetes self-management concern (AOR = 1.94, 95% CI: 1.04–3.61). Hyperglycemia in type 2 DM was increased by age (OR = 1.02, 95% CI: 1.00–1.04); high self-management concern (AOR = 2.59, 95% CI: 1.74–3.84); and poor confidence in self-management capacity (AOR = 3.22, 2.07–5.00). Conversely, diabetes app use (AOR = 0.63, 95% CI: 0.41–0.96) and “general diet” self-care (AOR = 0.84, 95% CI: 0.75–0.94), were significantly associated with the reduced odds of hyperglycemia. **Conclusion:** Diabetes apps, CGM, and educational interventions aimed at reducing self-management concerns and enhancing dietary self-care behavior and self-management confidence may help patients with diabetes to improve glycemic control.

## 1. Introduction

In 2017, more than 425 million adults were reported to be living with diabetes mellitus (DM). This estimate is expected to rise to 629 million cases by 2045 [1].

Effective DM self-management is often a challenging responsibility [2,3,4] which requires regular monitoring of blood glucose levels, adherence to glucose lowering medications, regular physical activity, and adhering to healthy nutrition recommendations [3,5,6,7]. Evidence suggests that receiving psychological and social support from other people affected by DM may help persons with DM tackle these challenges and enhance self-management capabilities [8]. People with similar conditions tend to connect with each other to learn from each other’s experiences. Social media, such as Facebook and disease-specific forums, may help to facilitate connections among persons with DM. In disease-specific Facebook groups and online forums, persons with DM can receive virtual support from fellow patients with similar conditions [9,10]. Therefore, the use of Facebook groups and other social media may be associated with improved knowledge regarding disease management, increased skills, and an increased likelihood for adopting the recommended lifestyle changes [11].

Facebook, having more than 2 billion users [12], has a special feature known as the “Facebook Group” to engage people with similar interests or similar health conditions [13]. These groups help patients exchange information through forming public, closed, or secret Facebook groups [14]. Facebook users have used this opportunity and many chronic disease specific Facebook groups have emerged over the past few years. Disease-specific Facebook groups and online forums are becoming popular among patients with chronic conditions. These new communication platforms may facilitate the exchange of experiences and support between people with similar problems [15].To date, diabetes is one of the most common chronic diseases by which Facebook groups are popular [11,13,16,17]. In recent studies, improvements in glycemic values, disease management capacity, and peer support as a result of participation in diabetes Facebook groups were reported [11,18].

Engaging in Facebook groups may help people with diabetes improve their knowledge of the disease and to help them learn more about the latest technological developments in diabetes, such as diabetes smartphone applications (apps). Diabetes smartphone apps aiming to support diabetes self-management are widely available [19,20]. Self-management responsibilities, such as blood glucose monitoring, adherence to insulin and dose calculations, tracking physical activity and nutrition can effectively be supported using diabetes apps [19,21,22,23]. Results of three studies suggest that the tracking of self-monitoring of blood glucose levels is the most common used functionality of diabetes apps [19,22,24]. A plethora of evidence also suggests that active use of diabetes and self-management apps plays an important role in improving clinical and behavioral outcomes of DM [25,26,27,28]. Findings of systematic reviews and meta-analyses of randomized-controlled trials suggested that diabetes apps and digital interventions led to reductions in glycated hemoglobin levels [27,29,30].

Supporting diabetes self-management efforts with mobile apps has become a cost-effective strategy due to the ubiquitous, multi-tasking, and easily portable nature of smart phones. Hence, there is a growing interest in using diabetes apps for diabetes care. In 2016, approximately 1800 diabetes apps were available for patients and providers. The global diabetes app use statistics have also increased from 2.2% in 2014 to 3.3 % in 2016 [31]. Of the more than four hundred million people living with DM, 135.5 million are potentially reachable via diabetes apps [31]. In addition, a survey with 500 app developers found that the biggest app market was targeted towards diabetes [32]. For this reason, diabetes has been perceived as a particular niche for smartphone health app innovation and marketing [33,34,35,36].

Recently, the International Diabetes Federation-Europe stated that well-suited diabetes apps might be worth a try, since it could transform mobile phones into medical devices promoting self-management practices, preventing complications, and improving the quality of life [37,38]. Although diabetes apps have a great potential for improving diabetes outcomes, concerns are also being raised regarding gaps between evidence-based recommendations and the app functionalities, the lack of integration between the apps and the healthcare delivery system, data privacy, clinical appropriateness, and safety [19,24,35,39,40]. In 2013, only one of 600 apps available in the United States received the Food and Drug Administration (FDA) approval [35,41]. The majority of these apps, particularly insulin dose calculator apps, were also reported to be erroneous. In fact, only one out of the total of 46 insulin dose calculator apps were found to be clinically suitable [42]. A recent report (May 2018) indicated that only 11 self-management apps were studied for clinical effectiveness, and only five were found to be clinically significant in terms of affecting HbA1c-levels [43]. For this reason, health care providers are required to exercise substantial caution and to systematically evaluate diabetes apps before prescribing them to their patients [42]. In addition, the need for formal evaluation and review remain essential [35,44].

Studies mainly stemming from developed countries indicate that a high proportion of people with DM own a smartphone and diabetes mobile apps. In Australia, 21% of patients with type 1 DM were reported to be using diabetes apps [45]. In New Zealand, approximately 20% of people with DM were using diabetes apps [46]. Survey results from Germany indicated that more than half of the respondents (60%) were using health apps, of which 84% were diabetes apps. Nearly 80% of the respondents in the German survey reported believing that diabetes apps were helpful to better cope with diabetes [47].

The American Food and Drug Administration Authority (FDA) developed classification regulations to evaluate and approve diabetes apps [48]. While there are ongoing efforts to clinically evaluate and approve diabetes apps, it is still unclear whether certified or other diabetes apps are being used and by whom, and which proportion of patients are using these apps. Furthermore, the relationship between use and glycemic control, diabetes self-management, and self-care behavior remain unclear. Additionally, we are unaware of evidence on the use of diabetes apps among persons with DM who subscribed to Facebook groups and other online forums. Therefore, in this study, we aimed at investigating the determinants (with special emphasis on the role of continuous glucose monitoring (CGM) device use, diabetes app use, and self-care behavior) of glycemic control among the online community of patients with type 1 and type 2 diabetes.

## 2. Materials and Methods

### 2.1. Ethics Approval and Consent to Participate

The content and implementation of the survey adheres with the overall ethical guidelines of the Leibniz Institute for Prevention Research and Epidemiology. In addition, the University of Bremen Central Research Development Fund committee approved the study and funded the cost of the Amazon vouchers. Before entering the survey, all participants were informed that the survey was voluntary, anonymous, and that they could skip any question that they were not comfortable to answer or could stop at any stage of the survey. Each participant was required to read and declare that he/she agreed or disagreed to answer questions of the survey. Participants were also required to electronically sign before their participation in the study. No personal data was collected. For the purpose of delivering incentives for the randomly selected participants, email addresses were collected in a separate database. However, no email addresses were linked to the responses of the participants. Email addresses were deleted after randomly selecting the participants for winning the draw.

### 2.2. Study Design, Questionnaire and Source of Respondents

From November 2017 to March 2018, we conducted a web-based cross-sectional survey among persons with diabetes using Facebook groups and diabetes-specific patient forums, and via targeted Facebook advertisements (ads). The survey questions were prepared in two languages (German and English) and were designed using Lime survey [49]. Self-reported diabetes status, demographic characteristics, type of diabetes, use of medication, self-care behavior, self-reported blood glucose level, use of continuous glucose monitoring (CGM), self-reported confidence in diabetes self-management, and perceived metabolic control were assessed in the survey. In addition, we asked about the use and names of diabetes smartphone apps by employing an adapted version of the mobile app rating scale [6], which has been used and validated in another diabetes app use survey conducted in New Zealand [46]. Self-care behavior was measured using a licensed version of the summary of diabetes self-care activities (SDSCA) scale. From previous studies, the SDSCA scale was evaluated for adequate reliability and was validated in the English [50] and German languages [51]. It included 11 questions measuring self-care activities related to diet, physical activity, blood glucose monitoring, foot care, and smoking [50].

### 2.3. Recruitment of Survey Participants

Survey participants were recruited via Facebook groups, targeted Facebook ads, and diabetes specific online forums.

#### 2.3.1. Recruitment via Facebook Groups

Facebook groups for type 1 or type 2 DM in English and German were systematically searched for on Facebook. Two Facebook accounts were used for the search. One of them was created for the purpose of the study. Before the search was conducted, the search histories of the two Facebook accounts were erased to avoid the impact of any previous searches. Key words, such as ‘diabetes’, ‘diabetic’, ‘type 1 diabetes’, ‘type 2 diabetes’, ‘support’, ‘group’, ‘diabetic’, and ‘diabetic friendly’ were used to search for relevant groups. Additional diabetes groups suggested by Facebook were also included. Groups for gestational DM, parents of children with DM, or owners of diabetic support pets were excluded. Two investigators M.M.K. and C.S. collected the group names, nature of the group (closed, public, and secret), URL, and the number of group members. After identification of the groups, two investigators (M.M.K. for the English groups and C.S. for the German groups) submitted applications to join the groups with explanations regarding the aims of the survey. Three Facebook accounts were used for requesting to join the groups and contacting the administrators (admins) of the groups. Admins were subsequently contacted to explain the authenticity and purpose of the survey, informed consent, and the amount of time necessary to complete the survey. Upon reception of approval, the survey link and an explanation of the purpose of the study, informed consent, and the time required to complete the survey was posted on each group page. Either an English or German version of the questionnaire was posted in the groups considering the language of communication of the groups. To make the post appear in the newsfeed of members and possibly enhance its visibility, it was periodically (at least every 24 h) bumped up by posting comments on the existing post. To encourage participation, the post included €50 Amazon voucher incentives to be given to randomly selected respondents.

#### 2.3.2. Recruitment via Targeted Facebook Ads

Facebook ads containing the survey URL were used to target potentially eligible people with DM by tailoring the advertisement to specific locations and interests. The ads were in English and German to target people with special interests and living in English and German speaking countries. The ads targeted persons who were 18 years and older living in Australia, Canada, the United Kingdom, the United States, Germany, Switzerland, and Austria and who had a special interest in diabetes-related Facebook pages using specific terms, such as “diabetes health”, “healthy low-carb living”, “cure for diabetes”, “glucose buddy”, “glycemic index”, “eating healthy food”, and “diabetic kitchen”. Additional terms suggested by Facebook and judged as relevant were also considered. The full list of terms used for the targeted ads can be accessed in the screenshot of one of our ads, presented in the Appendix A. We assumed that Facebook pages with these terms predominantly attracted users with DM or those at an increased risk for DM.

#### 2.3.3. Recruitment Using Diabetes Online Patient Forums

English and German diabetes patient forums were searched in Google using key words, such as ‘diabetes’, ‘forums’, ‘patient forums’, and ‘discussion forums.’ After collecting the names and URLs of the forums, two authors (M.M.K. and C.S.) registered or submitted registration requests to the forums. The two authors sent emails or personal messages to the forum admins and moderators to explain the purpose of the survey. After permission was received from the forum admins and moderators, the survey was posted on the diabetes forum website. The post also included information regarding the authenticity and purpose of the survey, informed consent, and the amount of time required to complete the survey.

### 2.4. Quality of Data

Periodically, the primary investigator checked the quality and authenticity of responses one by one. Multiple responses were removed using IP addresses. Incomplete responses were excluded from the final analysis.

### 2.5. Data Analysis

The data retrieved from lime survey were exported to MS Excel. STATA version 14 and R were used for analyses. Statistical packages such as “dplyr” and “ggplot2” packages in R were used to understand and prepare the data for analyses. The main outcome variable was the self-reported glycemic levels. Self-reported HbA1c and capillary blood glucose level data were collected. Respondents were asked to provide HbA1c values and to mention where the test was obtained. In addition, data on capillary blood glucose values and the timing of the test (pre-prandial, post-prandial, etc.) were also obtained. Moreover, data regarding how frequent respondents had hyperglycemia or hypoglycemia were collected. The combined self-reported HbA1c and capillary blood glucose data corresponding to the timing of the measure (pre-prandial, post-prandial) was assessed. The self-reported glycemic level data was then classified into three categories: hypoglycemia, hyperglycemia, and good glycemic control, following the American Diabetes Association guideline. Hence, self-reported HbA1c-levels of <7.0% or <53 mmol/mol, or a pre-prandial capillary plasma glucose levels between 80–130 mg/dL or 4.4–4.7 mmol/L, or post-prandial capillary glucose levels <180 mg/dL or <10.0 mmol/L were considered as good glycemic control levels. Self-reported HbA1c-levels of >7.0% or >53 mmol/mol, or a pre-prandial capillary plasma glucose level >130 mg/dL or >4.7 mmol/L, or post-prandial capillary glucose levels >180 mg/dL or >10.0 mmol/L were classified as hyperglycemia. HbA1c-levels reported as ≤70mg/dL or ≤3.9mmol/L were categorized as hypoglycemia [52,53,54].

Descriptive statistics and multinomial logistic regression analyses were performed. Associations of the independent variables with glycemic control were assessed using multinomial logistic regression analyses stratified by the type of diabetes. Variables were entered step-by-step in the model. The variables entered in the model included age, sex, and educational status, use of glucose lowering medication, use of a CGM device, self-care behavior, smoking status, diabetes self-management concern, and diabetes app use. Self-care behavior data were analyzed according to the recommendation provided in the summary of diabetes self-care activities (SDSCA) questionnaire [50,52]. Accordingly, the mean number of self-care days was calculated for all the self-care activities (i.e., “general diet”, “specific diet”, “exercise”, “blood glucose testing”, and “foot care”). Diabetes self-management concern was measured by aggregating the total responses of eight “yes” or “no” questions, which were coded as 1 and 0, respectively. These questions included concerns about hypoglycemia, hyperglycemia, forgetting to measure blood glucose levels, forgetting to take medications, not knowing whom to contact in case of a need for assistance, being left out of medication or supplies, and feeling unsure about how to calculate insulin doses. The total number of diabetes self-management concerns was calculated for each respondent. Checking for normal distribution using the Shapiro–Wilk test, we found that diabetes self-management concern was not normally distributed. Hence, respondents having more than or equal to the median number of diabetes self-management concerns (median = 3) were coded as having a “high concern” otherwise it was coded as a “low concern”.

Country level sub-group analyses were conducted to investigate whether the factors associated with glycemic level differed across countries. We conducted the sub-group analyses for respondents from the US, the UK, and Germany. Odds ratios with 95% confidence intervals and *p*-values of less than 0.05 were used to declare statistically significant association. The goodness of fit of the model was investigated using the McKelvey & Zavoina Pseudo-R2 method [55]. Stata 14 (StataCorp LP., Texas, TX, USA) version 14 and R studio (RStudio, Inc., Boston, MA, USA) were used for the analyses.

## 3. Results

### 3.1. Data Source and Characteristics of the Survey Participants

A total of 117 Facebook groups were identified through key-words searching. Contacting admins and moderators, as well as group joining requests, were submitted to all Facebook groups identified through the key-words search. The majority of the Facebook group joining requests (*n* = 98 (84%), were approved. Of the 98 Facebook groups which approved the group requests, more than three-fourths (*n* = 78, 80%) of the Facebook groups approved the survey URL to be posted on the group page (Figure 1).

A total of ten targeted Facebook ads were published on Facebook reaching about 30,000 people potentially having diabetes. In addition, 14 diabetes patient online forums were identified, and the survey URL was posted on more than half of the forums (Figure 1).

In total, 1854 complete and 352 incomplete responses were obtained. Of the 1854 persons with complete questionnaires, 1753 persons with diabetes reported owning a smartphone. Of those who owned a smartphone, 1682 respondents were persons with type 1 DM (*n* = 1052, 62.6%) or type 2 DM (*n* = 630, 37.4%) (Figure 1).

The mean age of the respondents with type 1 DM was 39 (SD = ±12.9) years, while it was 52.9 (SD = ±11.4) years for respondents with type 2 DM. More than two-thirds of the respondents in both type 1 and type 2 were females (763, 72.5% vs. 420, 66.7%, respectively).

The respondents were from 62 countries. However, according to the World Bank 2017–2018 country classifications [56] nearly all of the respondents were from high income countries (*n* = 1557, 92.3%) (Table 1). More than two-thirds of the respondents were from three countries: The United States (*n* = 543, 32.8%), Germany (*n* = 385, 22.9%), and the United Kingdom (*n* = 224, 13.3%).

### 3.2. Diabetes Clinical and Self-Management Characteristics of Respondents

The majority of the respondents, both with type 1 DM and type 2 DM reported taking glucose lowering medications (1004 (95.4%) vs. 541(85.9%)). About 4.6% of the respondents with type 1 diabetes did not report taking glucose-lowering medications or data about their treatment history were not available. A quarter of the respondents with type 1 DM and more than one third of the respondents with type 2 DM had hyperglycemia. More than a quarter of the respondents with type 1 DM reported using CGM technology (*n* = 296, 28.1%). More than half of the respondents with type 1 DM (*n* = 655, 62.4%) or type 2 (*n* = 323, 51%) rated their metabolic control as “well-controlled.” In addition, more than two-thirds of the respondents with type 1 DM (*n* = 706, 67.2%) rated their confidence in diabetes self-management as “very confident”, compared to less than half of the respondents with type 2 DM (*n* = 282, 44.8%) who felt “very confident”. More than half of the respondents with type 1 diabetes (*n* = 549, 52.2%) and more than one third of the respondents with type 2 diabetes (*n* = 210, 33.3%) reported using diabetes apps. More than a quarter of the respondents with type 1 or type 2 DM reported they first consulted Facebook groups, the internet, or diabetes apps, whenever they had concerns regarding their diabetes self-management (Table 2). 

### 3.3. Self-Care Behavior of the Respondents

The mean (SD) for self-care activity days spent per week on general diet was 4.5 (±2.01) amongst the respondents with type 1 DM and 4.7 (±1.92) amongst those with type 2 DM. The mean number of days per week for blood glucose testing self-care activity was 6.3 (1.47) among persons with type 1 diabetes, while it was 4.6 (2.58) among respondents with type 2 diabetes. Compared to the respondents with type 2 DM, respondents with type 1 DM reported a significantly higher mean number of days for blood glucose testing (difference = 1.7, two sample *t*-test *p*-value < 0.001). Furthermore, more self-care days were spent on foot care among respondents with type 1 DM compared to those with type 2 DM (difference = 0.8, two sample *t*-test value < 0.001) (Table 3).

Amongst the respondents with type 1 DM, for the self-care activity ‘blood glucose monitoring’, the highest mean number of days was reported. Respondents with type 1 DM with good glycemic control, hyperglycemia, or hypoglycemia reported 6.5 (±1.3), 5.9 (±1.8), and 6.2 (±1.8) days of blood glucose testing per week, respectively. However, persons with type 2 DM reported the highest number of self-care days for ‘general diet’ followed by ‘blood glucose monitoring’ (Table 4).

Both the persons with type 1 and type 2 DM who reported good glycemic control, also reported a nearly similar mean number of days spent on each self-care activity, except for foot care. Respondents with hyperglycemia or hypoglycemia reported generally fewer days spent on self-care compared to respondents with well-controlled type 1 or type 2 DM (Figure 2).

### 3.4. Factors Associated with Hyperglycemia and Hypoglycemia amongst Respondents with Type 1 Diabetes

Amongst the respondents with type 1 DM, educational status, smoking, CGM use, diabetes self-management concern, and only two of the self-care activities (general diet and blood glucose monitoring), were significantly associated with the odds of having hyperglycemia. Compared to the respondents with type 1 DM having a bachelor’s degree and above, respondents with a primary to secondary school educational level were 1.7 times (95% CI: 1.21–2.39) more likely to have hyperglycemia. Similarly, respondents who were smokers and had reported having a high diabetes self-management concern were 1.63 (95% CI: 1.15–2.32) and 2.09 (95% CI: 1.15–2.32) times more likely to have hyperglycemia, respectively. Respondents who were using CGM technology were 34% less likely to have hyperglycemia. In addition, a one-point increase on the scale for “general diet” self-care behavior and “blood glucose monitoring” reduced the odds of hyperglycemia by 14% and 12%, respectively (Table 5).

However, only CGM use and diabetes self-management concern were significantly associated with hypoglycemia among respondents with type 1 diabetes. The odds of having hypoglycemia were 76% lower amongst respondents who were using CGM technology. Compared to respondents having low concern in their diabetes self-management, respondents with a high concern were 1.94 times (95% CI: 1.04–3.61) more likely to experience hypoglycemia (Table 5).

### 3.5. Factors Associated with Hyperglycemia and Hypoglycemia Among Respondents with Type 2 DM

Age, diabetes app use, general diet, diabetes self-management concern, and self-reported confidence regarding diabetes self-management were significantly associated with experiencing hyperglycemia amongst respondents with type 2 DM.

Respondents who were using diabetes apps were 37% less likely to experience hyperglycemia than those who did not. A one-point increase on the “general diet” self-care scale was associated with a 16% reduction of the odds of experiencing hyperglycemia.

Respondents with type 2 DM that reported a high concern regarding their diabetes self-management were 2.59 (95% CI: 1.74–3.84) times more likely to experience hyperglycemia than those who had a low concern. Moreover, respondents who were not confident in their diabetes self-management capacity were 3.22 (95% CI: 2.07–5.00) times more likely to experience hyperglycemia than those who were confident. For each year increase in age, the odds of having hyperglycemia increased by a factor of 1.02 (95% CI: 1.00–1.04) (Table 5).

Regarding hypoglycemia among respondents with type 2 DM, the likelihood of hypoglycemia increased by a factor of 1.07 (95% CI: 1.01–1.14) for each additional increase in age. In addition, respondents that had a polytechnic diploma were 11(1.06–113.9) times more likely to report hyperglycemia compared to respondents who had a bachelor’s degree and above (Table 5).

Stratified analyses to investigate the country-level differences between the US, Germany, and the UK, regarding the factors associated with hyperglycemia and hypoglycemia among persons with type 1 and type 2 diabetes were conducted. The results were generally consistent with the pooled analyses. The results revealed that high diabetes self-management concern was consistently associated with increasing the odds of experiencing hyperglycemia amongst respondents with Type 1 diabetes who were from the US and Germany. Conversely, an increment in the blood glucose monitoring self-care days was consistently associated with reducing the odds of experiencing hyperglycemia amongst US and German respondents living with type 1 diabetes (Appendix A). However, for US respondents with type 1 diabetes, an increment in the physical activity self-care days was significantly associated with reducing the odds of experiencing hyperglycemia (Appendix A). For respondents with type 1 diabetes who were from the UK, the odds of experiencing hyperglycemia were significantly reduced, particularly by using CGM and increments in the specific diet self-care days (Appendix A).

Among respondents with type 2 diabetes who were from Germany or the US, an increment in the general diet self-care behavior days was significantly associated with reducing the odds of experiencing hyperglycemia in type 2 diabetes. However, having a high diabetes self-management concern and being in older age groups consistently increased the odds of experiencing hyperglycemia for both the respondents from the US and Germany (Appendix A)

## 4. Discussion

This was the first multi-national study aimed at investigating the role of CGM, diabetes app use, and self-care behavior in reducing glycemic control amongst persons with type 1 and type 2 DM. The study revealed that more than half of the respondents with type 1 diabetes and more than one third of the respondents with type 2 diabetes reported using diabetes smartphone apps to assist them in their disease self-management. Diabetes app use in this study was higher than reported from other studies. For example, a study conducted in New Zealand reported a use of 20% among people with type 1 diabetes [46], another study conducted in Australia reported user rates of 24% for persons with type 1 DM and 8% for persons with type 2 DM [57]. It was also much higher than user rates reported in studies from the US (4%) [22] and Scotland (7%) [58]. This difference was possibly due to the difference in the digital literacy among respondents [59,60,61]. People who use social media have a higher level of digital literacy. The relatively higher proportion of diabetes app use among people with type 1 diabetes was due to their younger age because of an earlier onset of the disease. Younger adults have a higher level of digital literacy. This is in line with other studies which found higher levels of app use in younger populations [57,62,63]. Possibly, it might also be due to the improved awareness of the consequences of failing to monitor blood glucose levels.

On average, respondents of this study reported spending more than two days per week on almost all self-care activities. Blood glucose monitoring and general diet were the two most commonly practiced self-care behaviors in both groups of respondents. Similarly, results by Tricia et.al indicated that patients with type 2 DM in the US spent more days on healthy eating and self-monitoring of blood glucose than on any other self-care activities [64]. However, whether there is a difference in the self-care behavior among diabetes app users and non-users who have type 1 or type 2 DM needs further investigation.

Having a lower educational status, smoking, and having a higher diabetes self-management concern increased the odds of experiencing hyperglycemia among respondents with type 1 DM. Previous studies reported a similar relationship between lower educational status and hyperglycemia [65,66]. This can partly be explained by the lower educational status which may inhibit the success of an interactive oral and written communication, which are key aspects of a successful patient–provider relationship and diabetes self-management [67,68,69]. In addition, education is an established mechanism that affects health literacy, where persons with higher levels of education may also have a greater diabetes self-management capacity and may feel more confident in achieving metabolic control [70].

Similar to the findings of our study, a plethora of evidence suggests that smoking increases the risk of hyperglycemia in persons with type 1 DM [66,71,72,73,74]. Smoking induced hyperglycemia, as explained in previous studies, might be due to the acute biochemical reactions caused by smoking, particularly the mobilization of catecholamines and increased cortisol productions [71]. In addition, smoking increases the risk of hyperglycemia by inhibiting insulin sensitivity and resistance [75,76].

Encouragingly, two self-care behaviors, namely “general diet” and “blood glucose monitoring”, were significantly associated with the reduced odds of experiencing hyperglycemia. Similarly, Schmitt and colleagues reported a higher number of self-care days for “general diet” and “blood glucose monitoring”, which were in turn correlated with glycemic control [77]. A generalized diet plan, regardless of specific calorie levels, with consistent carbohydrate intake is considered apractical methods of serving food while potentially improving glycemic control [78].

Interestingly, the use of CGM technology for glucose monitoring was inversely associated with both hyperglycemia and hypoglycemia among respondents with type 1 DM. In comparison, in numerous other experimental and observational studies, the use of CGM was associated with improved glycemic control [79,80,81]. CGM plays an important role in successful glycemic control by helping patients to timely and effectively detect and counteract hyperglycemia and hypoglycemia. It is considered as an optimal way to respond to glycemic abnormalities and glycemic variabilities in a timely manner [80,81]. However, CGM is more effective when combined with behavioral and educational strategies to respond to abnormal glycemic readings through the appropriate corrective actions, such as adjusting medication doses and changes to physical activity and nutrition [80,82].

Having a higher diabetes self-management concern increased the odds of experiencing hyperglycemia or hypoglycemia in persons with type 1 DM. This might be due to the poor self-management capacity, lack of problem-solving skills, and poor capacity of how to timely detect and appropriately respond to any episode of glycemic abnormalities [83,84,85]. Due to the frequent requirements of self-monitoring of blood glucose and other highly demanding self-management responsibilities, as well as lack of sleep, and worries about glycemic abnormalities, patients with type 1 diabetes have an increased risk of depression, anxiety, and emotional stress [86]. This may also increase the risk of experiencing hyperglycemia and hypoglycemia [87].

Looking at the factors associated with hyperglycemia among respondents with type 2 diabetes, it shows that diabetes app use and “general diet” appear to be two of the key factors to lowering the odds of having hyperglycemia. On the other hand, higher diabetes self-management concern and lower self-reported confidence with regard to diabetes self-management increased the odds of experiencing hyperglycemia.

Similar to the results reported in the current study, previous studies have reported lower risks of hyperglycemia among diabetes smartphone app users with type 2 diabetes [88,89]. Numerous randomized controlled trials also demonstrated the benefits of using diabetes smartphone apps for improving glycemic control in patients with type 2 DM [21,90,91]. Diabetes apps help improve glycemic control by enhancing diabetes knowledge, self-management capabilities, adherence to medication, and healthy life style recommendations [19,21,22,23,24,92]. Diabetes apps have the potential to facilitate personalized medicine by helping patients with type 2 diabetes to achieve their personalized self-management goals and improve clinical and behavioral outcomes. However, the quality and content of the apps requires further research to help patients and providers to identify effective and user-friendly diabetes apps. In addition, considering the large number of diabetes apps currently available online, choosing the right app might be difficult for patients. Hence, the evaluation and recommendation of physicians are important before using them. Further research is also required to identify the most popular diabetes apps.

In the current study, the odds of experiencing hyperglycemia were reduced with increments in the “general diet” self-care behavior. Related to this finding, previous studies also reported that poor compliance to a healthy diet was a significant predictor of hyperglycemia [93,94,95,96]. Consistent meal planning, particularly to maintain the consistency of carbohydrate intake, has been an accepted standard for improving glycemic control [78]. However, whether low or high carbohydrate meals are preferable for improving glycemic control is controversial and there is still conflicting evidence on the issue [97,98,99,100].

Although the association was not strong, each additional year increase in age significantly increased the odds of having both hyperglycemia and hypoglycemia among respondents with type 2 diabetes. Diabetes is considered a progressive disease and glycemic levels worsen with increasing age [6,101]. This is partly due to the decline in beta-cell function, impaired insulin secretion, as well as lack of metformin effectiveness among the elderly patients [101,102,103,104].

Similar to patients with type 1 diabetes, patients with type 2 DM reporting a high diabetes self-management concern were more likely to have hyperglycemia. This might be due to their lack of diabetes knowledge, and psychosocial and emotional challenges [105] affecting self-management skills and resilience to glycemic abnormalities [83,84,85].

Patients with low self-rated confidence in their diabetes self-management management capacity were more likely to experience hyperglycemia compared to patients who rated their confidence as high. In line with this finding, a study by Whittemore and colleagues reported diabetes self-management confidence as one of the most consistent predictors of metabolic control [106]. Educational interventions aiming to enhance confidence in the self-management capacity of patients may help patients to attain their personalized glycemic control goals.

### Limitations

This study had several limitations. All the respondents of the survey were recruited using online platforms. Hence, patients who do not subscribe to Facebook and diabetes online forums were not represented in this study, which limits the generalizability of the study. In addition, participants of the study came from multiple countries. The impact of a broad range of factors, such as differences in the healthcare systems, and social, cultural, and racial differences were not investigated in the study. A further stratified analyses of the data based on the nationality or income category of respondents may provide a better insight and help minimize the bias due to unobserved variation arising from the health care systems, as well as socio-economic and cultural differences among respondents. These factors potentially limit the applicability of the results to a specific context. However, patients with diabetes who are subscribers to social media are also an important population presenting an interesting paradigm of self-management and social support. Therefore, considering the ever-increasing presence and interest of patients with diabetes on social media, who are seeking diabetes-related information and support, this particular community of diabetes requires attention. In this regard, our research attempted to investigate glycemic control in this particular population. Moreover, the responses to all questions, including biochemical parameters, were based on self-reported responses and were not objectively measured. Social desirability may have played a role while completing the survey. This may also have an impact on our results.

In addition, due to the cross-sectional nature of the study, the results should be carefully interpreted because HbA1c and one-time capillary glucose levels may not be adequate to understand the hyperglycaemic and hypoglycaemic episodes of patients with diabetes. Measuring several glucose levels per day is more helpful to understand the dynamics of glycemic control. Recent studies demonstrated that determining the coefficient of glycemic variation (CV) is an important metric to assess glycemic variability rather than using HbA1c. Ideally, this is only possible with access to CGM.

We noted that conducting a survey using Facebook groups was very challenging. Importantly, the members of these groups came from many parts of the world, which was very demanding because the researchers needed to work with multiple time zones. Keeping the survey active on the newsfeed of members required an extended effort. We did this by trying to engage and motivate members of the group to enter comments to the survey posted on the Facebook groups, and by periodically bumping the post so that members of a group who did not see the survey post may see it and participate in the survey. This was not as effective as we thought, especially in groups with tens of thousands of members. In larger Facebook groups that had a very active flow of information, the information posted on the group page vanished from the newsfeed in just a matter of a few minutes, unless the members entered comments or influential group admins were engaged to keep it active on the page. Hence, we advise future researchers to consider actively involving group champions (admins and moderators) in their research. Group champions, being the most influential members of Facebook groups, enhance the participation and response rates by motivating members, triggering discussions, and most importantly pinning the survey on a group page for a particular period of time.

## 5. Conclusions

Diabetes app use reduces the odds of experiencing hyperglycemia in type 2 DM. The use of CGM technology for monitoring blood glucose significantly reduces the odds of experiencing hyperglycemia and hypoglycemia in type 1 DM. Diabetes apps and the use of CGM may facilitate personalized medicine to helping patients achieve individualized glycemic goals. Educational interventions targeted at reducing self-management concern, improving dietary self-care behavior, and self-management confidence may help patients with type1 and type 2 DM reduce glycemic abnormalities.

## Figures and Tables

**Figure 1 jcm-08-00109-f001:**
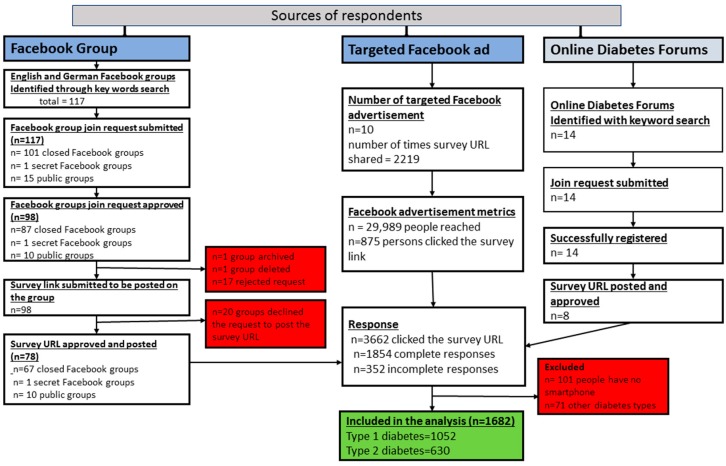
Flow diagram for the sources of respondents of the survey.

**Figure 2 jcm-08-00109-f002:**
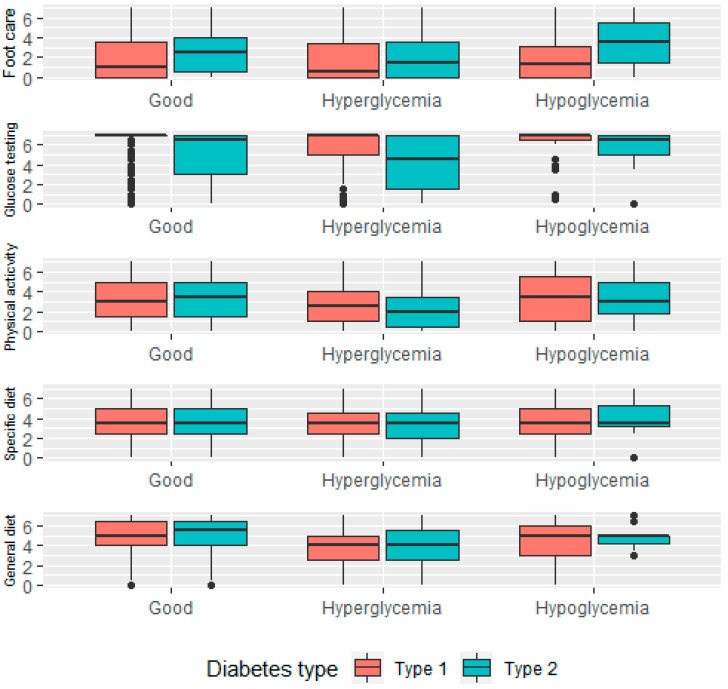
Self-care behavior distribution by glycemic levels.

**Table 1 jcm-08-00109-t001:** Characteristics of the respondents.

Variable	Respondents with Type 1 DM Glycemic Control Levels *N* (%)				Respondents with Type 2 DM Glycemic Control Levels *N* (%)			
	Good	Hyper	Hypo	Total	Good	Hyper	Hypo	Total
**Age, Mean (SD)**	40 (12.9)	36.7 (12.5)	36 (13.2)	39 (12.9)	52.8 (11.4)	52.8 (11.1)	61.5 (14.1)	52.9 (11.4)
≤40	379 (52.8)	178 (64)	34 (60.7)	591 (56.2)	70 (17.8)	29 (12.8)	0 (0)	99 (15.7)
40–60	294 (40.9)	86 (31)	20 (35.7)	400 (38)	203 (51.7)	137 (60.6)	6 (54.6)	346 (54.9)
60+	45 (6.3)	14 (5)	2 (3.6)	61 (5.8)	120 (30.5)	60 (26.6)	5 (45.4)	185 (29.4)
**Sex**								
Female	509 (70.9)	215 (77.3)	39 (69.6)	763 (72.5)	255 (64.9)	156 (69)	9 (81.8)	420 (66.7)
Male	209 (29.1)	63 (22.3)	17 (30.4)	289 (27.5)	138 (35.1)	70 (31)	2 (18.2)	210 (33.3)
**Educational Status**								
Primary to secondary	252 (35.1)	141 (50.7)	17 (30.4)	410 (39)	156 (39.7)	116 (51.3)	6 (54.5)	278 (44.1)
Polytechnic diploma	121 (16.9)	51 (18.4)	12 (21.4)	184 (17.5)	76 (19.3)	37 (16.4)	4 (36.4)	117 (18.6)
Bachelor degree and above	345 (48)	86 (30.9)	27 (48.2)	458 (43.5)	161 (40.1)	73 (32.3)	1 (9.1)	235 (37.3)
**Continent**								
USA/Canada/Central America	237 (33)	96 (34.5)	20 (35.7)	353 (33.6)	181 (46.1)	93 (41.2)	2 (43.8)	276 (43.8)
Europe	418 (58.2)	161 (58)	28 (50)	607 (55.7)	143 (36.4)	90 (39.8)	6 (55)	239 (38)
Oceania	49 (5.4)	8 (2.9)	5 (8.9)	52 (4.9)	18 (4.6)	4 (1.8)	2 (18.1)	24 (3.8)
Asia	10 (1.4)	4 (1.4)	1 (1.8)	15 (1.4)	41 (10.4)	25 (11)	1 (9.1)	67 (10.6)
Africa and Latin America	14 (2)	9 (3.2)	2 (3.6)	25 (2.4)	10 (2.5)	14 (6.2)	0 (0)	24 (3.8)
**Country income levels** *								
Low to lower-middle income	8 (1.1)	2 (0.7)	0 (0)	19 (1)	41 (10.4)	26 (11.5)	1 (9.1)	68 (10.7)
Upper-middle income	16 (2.2)	12 (4.3)	2 (3.6)	30 (3)	9 (2.3)	13 (5.8)	0 (0)	22 (3.6)
High income	694 (96.7)	264 (95)	54 (96.4)	1012 (96)	343 (87.3)	187 (82.7)	10 (90.9)	540 (85.7)
Total	718 (68.2)	278 (26.4)	56 (5.3)	1052 (100)	393 (62.4)	226 (35.8)	11 (1.8)	630 (100)

* Based on the World Bank 2017–2018 country classifications [56].

**Table 2 jcm-08-00109-t002:** Clinical and self-management characteristics of the respondents.

	Respondents with Type 1 DM Glycemic Control Levels *N* (%)				Respondents with Type 2 DM Glycemic Control Levels *N* (%)			
	Good	Hyper	Hypo	Total	Good	Hyper	Hypo	Total
**On glucose lowering medication**								
Yes	684 (95.3)	266 (95.7)	54 (96.4)	1004 (95.4)	332 (84.5)	202 (89.4)	7 (63.6)	541 (85.9)
No	34 (4.7)	12 (4.3)	2 (3.6)	48 (4.6)	61 (15.5)	24 (10.6)	4 (36.4)	89 (14.1)
**If you have concerns regarding your diabetes management where do you go first for assistance?**								
Diabetes specialist team/healthcare provider	445 (62)	180 (64.8)	35 (62.5)	660 (62.7)	265 (67.4)	156 (69)	10 (90.9)	431 (68.4)
Facebook group/Internet/Smartphone App	214 (29.8)	85 (30.6)	17 (30.4)	316 (30)	98 (24.9)	54 (23.9)	1 (9.1)	153 (24.3)
Support group/Friends/Family	50 (7)	12 (4.3)	4 (7.1)	66 (6.3)	24 (6.1)	14 (6.2)	0 (0)	38 (6)
Other	9 (1.25)	1 (0.36)	0 (0)	10 (1)	6 (1.5)	2 (0.9)	0 (0)	8 (1.3)
**Problems with Diabetes Self-Management**								
**Feeling Symptomatic Low Blood Sugar**								
Yes	443 (61.7)	182 (65.5)	38 (67.9)	663 (63)	87 (22.1)	33 (14.6)	1 (9.1)	121 (19.2)
No	275 (38.3)	96 (34.5)	18 (32.1)	389 (37)	306 (77.9)	193 (85.4)	10 (90.9)	509 (80.8)
**Feeling Symptomatic High Blood Sugar**								
Yes	321 (44.7)	180 (64.8)	31 (55.4)	532 (50.6)	87 (22.1)	109 (48.2)	4 (36.4)	200 (31.8)
No	397 (35.3)	98 (35.3)	25 (44.6)	520 (49.4)	306 (77.9)	117 (51.8)	7 (65.6)	430 (68.2)
**Forgetting to Measure Blood Sugar Levels**								
Yes	121 (16.9)	113 (40.7)	13 (23.2)	247 (23.4)	82 (20.9)	92 (40.7)	1 (9.1)	175 (22.8)
No	597 (83.2)	165 (59.4)	43 (76.8)	805 (76.5)	311 (79)	134 (59.3)	10 (90.1)	455(72.2)
**Forgetting to Take Medication or Insulin**								
Yes	106 (14.8)	70 (25.2)	10 (17.9)	186 (17.7)	49 (12.5)	58 (25.7)	2 (18.2)	109 (17.3
No	612 (85.2)	208 (74.8)	46 (82.1)	866 (82.3)	344 (87.5)	168 (74.3)	9 (81.8)	521 (82.7)
**Not knowing how to identify high or low blood sugars**								
Yes	37 (5.2)	15 (5.4)	5 (8.9)	57 (5.4)	34 (8.7)	31 (13.7)	0 (0)	65 (10.3)
No	681 (94.9)	263 (94.6)	51 (91.1)	995 (94.6)	359 ((91.4)	195 (86.3)	11 (100)	565 (89.7)
**Not Knowing whom to Contact when in Need of Assistance**								
Yes	29 (4)	11 (4)	1 (1.8)	41 (3.9)	26 (6.6)	24 (10.6)	0 (0)	50 (7.9)
No	689 (96)	267 (96)	55 (98.2)	1011 (96.1)	367 (93.4)	302 (89.4)	11 (100)	580 (92.1)
**Being Left without Medication/Supplies**								
Yes	64 (8.9)	36 (13)	5 (9)	105 (10)	25 (6.4)	19 (8.4)	0 (0)	44 (7)
No	624 (91.1)	242 (87)	51 (91)	947 (90)	368 (93.6)	207 (91.6)	11 (100)	586 (93)
**Felt Unsure about How to Calculate Your Insulin/Glucose lowering Medication Dose**								
Yes	105 (14.6)	64 (23)	18 (32.1)	187 (17.8)	14 (3.6)	16 (8.4)	1 (9.1)	34 (5.4)
No	613 (85.4)	214 (77)	38 (79.9)	865 (82.2)	379 (96.4)	207 (91.6)	10 (90.9)	596 (94.6)
**Diabetes App Use**								
Yes	401 (55.9)	122 (43.9)	26 (46.4)	549 (52.2)	156 (39.7)	53 (23.5)	1 (9.1)	210 (33.3)
No	317 (44.2)	156 (56.1)	30 (53.4)	503 (47.8)	237 (60.3)	173 (76.6)	10 (90.9)	420 (66.7)
**Use CGM**								
Yes	234 (32.6)	56 (20.1)	6 (10.7)	296 (28.1)	17 (4.3)	4 (1.8)	0 (0)	218 (3.3)
NO	484 (67.4)	222 (79.9)	50 (89.3)	756 (71.9)	376 (95.7)	222 (98.2)	11 (100)	609 (96.7)
**Self-Reported Rating of Blood Glucose Control**								
Well controlled	521 (72.8)	107 (38.5)	27 (48)	655 (62.4)	264 (67.5)	52 (23)	7 (63.6)	323 (51)
Neutral	149 (20.8)	96 (34.5)	11 (20)	256 (24.4)	88 (22.5)	63 (27.9)	3 (27.3)	154 (25)
Poorly controlled	46 (6.4)	75 (27)	18 (32)	139 (13.2)	39 (10)	111 (49.1)	1 (9.1)	151 (24)
**Self-Reported Confidence on Diabetes Self-Management**								
Very confident	533 (74.3)	140 (50.5)	33 (58.9)	706 (67.2)	221 (56)	55 (24.3)	6 (54.6)	282(44.8)
Neutral	66 (9.2)	48 (17.3)	8(14.3)	122 (11.6)	54(14)	41 (18.1)	2 (18.2)	97(15.4)
Not confident at all	118 (26.5)	89 (32.1)	15 (26.8)	222 (21.1)	118 (36)	130 (57.5)	3 (27.3)	251(39.8)
**Smoking**								
Yes	120 (16.7)	90 (32.4)	15 (32.4)	225 (21.4)	60 (15.3)	48 (21.2)	1 (9.1)	109 (17.3)
No	598 (83.3)	188 (67.6)	41 (73.2)	827 (78.6)	333 (84.7)	178 (78.8)	10 (90.9)	521 (82.7)
Total	718 (100)	278 (100)	56 (100)	1052 (100)	393 (100)	226 (100)	11 (100)	630 (100)

**Table 3 jcm-08-00109-t003:** Self-care behavior characteristics.

Self-Care Behavior	Type 1 Diabetes Mean (SD)	Type 2 Diabetes Mean (SD)	Difference (*p* value)
**Diet**			
How many of the last SEVEN DAYS have you followed a healthful eating plan?	4.5 (2.13)	4.6 (2.11)	0.412
On average, over the past month, how many DAYS PER WEEK have you followed your eating plan?	4.6 (2.07)	4.8 (1.97)	0.067
On how many of the last SEVEN DAYS did you eat five or more servings of fruits and vegetables	3.7 (2.51)	3.6 (2.54)	0.345
On how many of the last SEVEN DAYS did you eat high fat foods such as red meat or full-fat dairy products?	3.7 (2.3)	3.4 (2.41)	0.070
General diet (aggregate)	4.5 (2.01)	4.7 (1.92)	0.153
Specific diet(aggregate)	3.5 (1.78)	3.6 (1.81)	0.608
**Physical activity**			
On how many of the last SEVEN DAYS did you participate in at least 30 min of physical activity? (Total minutes of continuous activity, including walking).	4.0 (2.35)	3.7 (2.48)	0.022 *
On how many of the last SEVEN DAYS did you participate in a specific exercise session (such as swimming, walking, biking) other than what you do around the house or as part of your work?	2.4 (2.34)	2.5 (2.56)	0.524
Physical activity (aggregate)	3.2 (2.09)	3.1 (2.26)	0.356
**Blood Glucose Monitoring**			
On how many of the last SEVEN DAYS did you test your blood sugar?	6.7 (1.15)	5.0 (2.63)	<0.001 *
On how many of the last SEVEN DAYS did you test your blood sugar the number of times recommended by your health care provider?	6.0 (2.09)	4.2 (2.95)	<0.001 *
Blood glucose monitoring (aggregate)	6.3 (1.47)	4.6 (2.58)	<0.001 *
**Foot Care**			
On how many of the last SEVEN DAYS did you check your feet?	2.6 (2.76)	3.6 (2.89)	<0.001 *
On how many of the last SEVEN DAYS did you inspect the inside of your shoes?	0.9 (1.96)	1.6 (2.56)	<0.001 *
Foot care (aggregate)	1.8 (2.04)	2.6 (2.38)	<0.001 *

* statistically significant.

**Table 4 jcm-08-00109-t004:** Glycemic control status by self-care behavior in type 1 and type 2 diabetes.

Self-Care Behavior	Respondents with Type 1 DM Glycemic Control Levels *N* (%)				Respondents with Type 2 DM Glycemic Control Levels *N* (%)			
	Good	Hyper	Hypo	Total	Good	Hyper	Hypo	Total
**General Diet (Mean(SD))**	4.8 (1.9)	3.8 (2.1)	4.5 (2.1)	4.6 (2.0)	5.1 (1.8)	3.9 (2.0)	4.8 (1.2)	4.7 (1.9)
**Specific Diet (Mean(SD))**	3.6 (1.8)	3.3 (1.7)	3.6 (1.9)	3.5 (1.8)	3.7 (1.8)	3.4 (1.8)	4 (1.9)	3.6 (1.8)
**Physical Activity (Mean(SD))**	3.4 (2.1)	2.7 (2.0)	3.5 (2.3)	3.2 (2.1)	3.4 (2.2)	2.5 (2.2)	3.5 (2.2)	3.1 (2.3)
**Blood Glucose Monitoring (Mean(SD))**	6.5 (1.3)	5.9 (1.8)	6.2 (1.8)	6.3 (1.5)	4.9 (2.5)	4.1 (2.6)	5.5 (2.1)	4.6 (2.6)
**Foot Care (Mean(SD))**	1.8 (2.1)	1.6 (1.9)	1.9 (2.2)	1.8 (2.0)	2.7 (2.4)	2.4 (2.4)	3.5 (2.5)	2.6 (2.4)

**Table 5 jcm-08-00109-t005:** Multinomial logistic regression model of glycemic control in type 1 and type 2 DM.

	Type 1 Diabetes (*n* = 1052)		Type 2 Diabetes (*n* = 630)	
Variables	Hyperglycemia vs. Good Glycemic Control	Hypoglycemia vs. Good Glycemic Control	Hyperglycemia vs. Good Glycemic Control	Hypoglycemia vs. Good Glycemic Control
	AOR (95% CI)	AOR (95% CI)	AOR (95% CI)	AOR (95% CI)
**Age group**				
≤40	1 (reference)	1 (reference)		
40–60	0.78 (0.56–1.10)	0.80 (0.43–1.48)		
60+	1.09 (0.54–2.18)	0.62 (0.14–2.81)		
**Age (continuous)**			1.02 (1.00–1.04) *	1.07 (1.01–1.14) *
**Sex**				
Female	1 (reference)	1 (reference)	1 (reference)	1 (reference)
Male	0.80 (0.56–1.15)	1.17 (0.63–2.19)	1.12 (0.74–1.67)	0.24 (0.04–1.34)
**Education**				
Primary to secondary school	1.70 (1.21–2.39) **	0.69 (0.36–1.34)	1.30 (0.85–1.98)	8.56 (0.88–83.31)
Poly technique diploma	1.47 (0.95–2.27)	1.07 (0.51–2.23)	0.84 (0.49–1.44)	11 (1.06–113.9) *
Bachelor degree and above	1 (reference)	1 (reference)	1 (reference)	1 (reference)
**Diabetes app use**				
Yes	0.98 (0.69–1.39)	1.19 (0.65–2.20)	0.63 (0.41– 0.96) *	0.13 (0.01–1.14)
No	1 (reference)	1 (reference)	1 (reference)	1 (reference)
**Self-care behavior **				
General diet	0.86 (0.79–0.94) **	0.93 (0.79–1.09)	0.84 (0.75–0.94) **	0.80 (0.51–1.23)
Specific diet	1.00 (0.91–1.10)	1.01 (0.85–1.20)	1.02 (0.91–1.14)	1.13 (0.75–1.70)
Physical activity	0.93 (0.86–1.01)	1.09 (0.95–1.26)	0.96 (0.87–1.05)	1.12 (0.80–1.58)
Blood glucose monitoring	0.88 (0.80–0.97) *	0.91 (0.76–1.10)	0.96 (0.88–1.03)	1.26 (0.92–1.72)
Foot care	1.00 (0.92–1.08)	1.03 (0.90–1.19)	0.97 (0.89–1.05)	1.11 (0.84–1.47)
**Smoking**				
Yes	1.63 (1.15–2.32) **	1.67 (0.86–3.25)	1.16 (0.70–1.90)	0.57 (0.06–5.09)
No	1 (reference)	1 (reference)	1 (reference)	1 (reference)
**Glucose lowering medication **				
Yes	1.25 (0.61–2.54)	1.45 (0.33–6.36)	0.93 (0.52–1.68)	0.27 (0.06–1.22)
No	1 (reference)	1 (reference)	1 (reference)	1 (reference)
**Diabetes self-management concern**				
High concern	2.09 (1.50–2.92) **	1.94 (1.04–3.61) *	2.59 (1.74–3.84) **	0.83 (0.16–4.39)
Low concern	1 (reference)	1(reference)	1 (reference)	1 (reference)
**Use CGM**				
Yes	0.66 (0.44–1.00) *	0.24 (0.09–0.60) **		
No	1	1		
**Self-reported confidence on diabetes self-management**				
Very confident	1 (reference)	1 (reference)	1 (reference)	1 (reference)
Neutral			2.13 (1.23–3.72) **	1.53 (0.24–10.00)
Not confident at all			3.22 (2.07–5.00) **	1.12 (0.21–6.01)

** *p* < 0.01, * *p* < 0.05.

## Data Availability

The datasets collected, used, and analyzed for the study can be obtained from the corresponding author on a reasonable request. The University of Bremen Central Research Development Fund committee funded the cost of the Amazon vouchers given to survey participants in a lottery.

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
