# Peer review of "The Role of Continuous Glucose Monitoring, Diabetes Smartphone Applications, and Self-Care Behavior in Glycemic Control: Results of a Multi-National Online Survey"

_jcm, 2019, doi:10.3390/jcm8010109_

Round 1
Reviewer 1 Report
COMMENTS FOR AUTHORS: Role of continuous glucose monitoring
Line 118: Was the study approved by an institutional review board, and was signed informed consent obtained (even if the “signature” was electronic).
Line 162: Was the use of Amazon voucher incentives included in the review by the institutional review board, assuming that there was such a review?
Line 194: Presumably the patients did not check their hemoglobin A1C at home, do the authors have information on how this data was obtained?
Line 214 ff: Since diabetes self management concern was not normally distributed, why did the authors not use standard non-parametric statistics rather than the arbitrary dichotomous assignment above 3 and below 3, and what was done if the answer was 3?
Table 1: How were the criteria for “good” , “hyper”, and “Hypo” obtained? For example, was it possible to be included in Table 1 if there was only 1 glycemic control measure, how about 2, 3, 4, etc etc. Without this information, Table 1 is uninterpretable.
General comment: A crucial factor for this paper is definition of glycemic control levels. The authors must painstakingly tell the reader exactly how they aggregated the self-reported data to characterize the subjects as glycemic control as good, hyper, or hypo. They must also describe how they handled the various numbers of reports which went into the management characterization.
Line 332: delete the editorialism “obviously”
Line 333: The higher use of diabetes apps in type 1 diabetes might just as well reflect the patients’ knowledge that failure to carefully monitor their blood glucose would have immediate health consequences, and not be age related.
Lines 363 ff: The authors need to better discuss the paradox of both hyper-and hypo-glycemia involved in the use of CGM technology. Might it be that CGM methodology inevitably involves swings between hyper and hypoglycemia through over and under correction of insulin delivery based on the most recent glucose levels.
Author Response
COMMENTS FOR AUTHORS: Role of continuous glucose monitoring…. jcm 401282
Line 118: Was the study approved by an institutional review board, and was signed informed consent obtained (even if the “signature” was electronic).
Response: Yes, the study was ethically approved internally by the Leibniz Institute for Prevention Research and Epidemiology. In addition, the University of Bremen Central Research Development Fund committee approved the study and funded the cost of the Amazon vouchers. This is clarified in the ethical approval subtitle of the manuscript
Line 162: Was the use of Amazon voucher incentives included in the review by the institutional review board, assuming that there was such a review?
Response: Yes, the 50€ Amazon vouchers we gave to 10 randomly selected participants was approved by Leibniz Institute for Prevention Research and Epidemiology and funded by the University of Bremen Central Research Development Fund
Line 194: Presumably the patients did not check their hemoglobin A1C at home, do the authors have information on how this data was obtained?
Response: Thanks for raising this point. We made it clear at this point. The Hba1C values were also self-reported and we asked our respondents where they obtained their HbA1c values. They are all obtained from their providers.
Line 214 ff: Since diabetes self management concern was not normally distributed, why did the authors not use standard non-parametric statistics rather than the arbitrary dichotomous assignment above 3 and below 3, and what was done if the answer was 3?
Response: Diabetes self-management concern was measured by 8 yes or no questions. After checking for normal distribution, we used median cut point. So if the concern was greater or equal to 3, it was considered as “high concern”. There are many was to dichotomise a composite variable such as based on normal distribution (mean/median split), experience from previous analyses, arbitrarily, with confirmatory factor analyses, judgement or arbitrarily. All of these methods have their own pros and cons. Checking normal distribution helps to understand whether median or mean is the center of the data. If the data is not normally distributed, the median is the choice that equally divides the data into two equal halves. In non-normally distributed data, a median split is a preferable choice. (http://quantpsy.org/pubs/rucker_mcshane_preacher_2015.pdf ). We followed this method. Previous research indicated arbitrary dichotomization should be avoided. (https://www.ncbi.nlm.nih.gov/pmc/articles/PMC5506131/). As mentioned before, the methods have their advantages and disadvantages. As Altman DC, 2006 (https://www.ncbi.nlm.nih.gov/pmc/articles/PMC1458573/) highlighted keeping the variables as numeric scores might be the better option. However, the numeric scores are not clinically meaningful. Therefore, choosing one of these methods should be based on theoretical and statistical consideration.
Table 1: How were the criteria for “good” , “hyper”, and “Hypo” obtained? For example, was it possible to be included in Table 1 if there was only 1 glycemic control measure, how about 2, 3, 4, etc etc. Without this information, Table 1 is uninterpretable.
Response: We followed a standard definition to classify glycemic control into three categories. Most of our respondents provided HbA1c values. We also received self-reported capillary glucose levels which the respondents tested themselves at home. For these values, we asked our respondents the timing of their test (pre-prandial or post-prandial). These data were used to categorize the glycemic control levels. Based on the ADA definition, Self-reported HbA1c-levels of < 7.0% or < 53 mmol/mol, or a pre-prandial capillary plasma glucose levels between 80-130 mg/dl or 4.4-4.7 mmol/l, or post-prandial capillary glucose levels<180 mg/dl or <10.0 l="" were="" considered="" as="" good="" glycemic="" control="" levels.="" self-reported="" hba1c-levels="" of="">7.0% or > 53 mmol/mol, or a pre-prandial capillary plasma glucose level >130 mg/dl or >4.7 mmol/l, or post-prandial capillary glucose levels >180 mg/dl or >10.0 mmol/l were classified as hyperglycemia. HbA1c-levels reported as ≤70mg/dl or ≤3.9mmol/l were categorized as hypoglycemia.
General comment: A crucial factor for this paper is definition of glycemic control levels. The authors must painstakingly tell the reader exactly how they aggregated the self-reported data to characterize the subjects as glycemic control as good, hyper, or hypo. They must also describe how they handled the various numbers of reports which went into the management characterization.
Response: As described above we followed the ADA standard definition to categorize the glycemic levels into the three categories. Respondents have provided us their HbA1 values and their capillary blood glucose values. Respondents were asked to provide HbA1c values and to mention where the test was obtained. In addition, data on capillary blood glucose values and the timing of the test (pre-prandial, post-prandial, etc) were also obtained. We also collected data about how frequent respondents have hyperglycemia or hypoglycaemia.
We agree measuring several episodes of hyperglycaemia and hypoglycaemia is more helpful to understand the glycemic control of patients with diabetes. Determining coefficient of glycemic variation(CV) is an important measure of glycemic variability. However, this is only possible with CGM. We did not have access to CGM data http://care.diabetesjournals.org/content/diacare/40/8/994.full.pdf. We included this point in the limitation section.
Line 332: delete the editorialism “obviously”
Response: Thank you. It is now deleted.
Line 333: The higher use of diabetes apps in type 1 diabetes might just as well reflect the patients’ knowledge that failure to carefully monitor their blood glucose would have immediate health consequences, and not be age related.
Response: Thank you for suggesting an alternative explanation to elaborate the higher diabetes app use among patients with type 1 diabetes. As you mentioned, it could be awareness of the consequences of not monitoring their blood glucose levels. We tried to find literature to support this argument and we did not find any. However, we added your explanation in the discussion.
Lines 363 ff: The authors need to better discuss the paradox of both hyper-and hypoglycaemia involved in the use of CGM technology. Might it be that CGM methodology inevitably involves swings between hyper and hypoglycaemia through over and under correction of insulin delivery based on the most recent glucose levels.
Response: Our results indicate, CGM helps to prevent both hyper and hypoglycaemia. Compared to those who do not use CGM, a lower prevalence of hyperglycaemia and hypoglycaemia was reported from respondents using CGM. CGM does not inevitably involve glycemic swings if patients timely and appropriately react to abnormal glycemic levels.
Reviewer 2 Report
Role of Continuous Glucose Monitoring, Diabetes Smartphone Applications and Self-Care behavior in Glycemic Control: Results of a Multi-national Online Survey – Kebede et al.
1. Summary
In this work, the authors report the results of a multi-national online survey investigating continuous glucose monitoring sensors, smartphone applications usage, and therapy behavior impact on patients affected by type 1 and type 2 diabetes. This study is motivating and offers many ideas to further investigate the topic of interest. Moreover, it is both curious and appealing how the authors used the social media to collect the data. However, I found some weak points that need attention.
2. Major Comments
1. Line 146: The author should verify and prove that, in Facebook, just clearing search histories avoid the impact of previous searches on future ones. In fact, my guess is that the “Clear searches” feature of Facebook is just a visualization tool to “clear” the search field from listing previous searches. Maybe, the best solution would have been creating two brand new accounts instead of using already existing ones.
2. Line 191: It is not clear how the target variable of the multinomial logistic regression has been defined.
3. Line 203: It would be desirable the authors add the description on how data have been prepared in order to train the multinomial logistic regression model.
4. Limitation section: the author should discuss about the fact that it is not possible to be 100% sure that engaging respondents from Facebook groups and online forums, they are all patients with type 1 and type 2 diabetes. Indeed, it could happen that fake accounts are recruited instead.
5. One of the major confounders in this analysis is the nationality of the respondents. Indeed, it is well-known that different countries have different dietary regime, culture, and average behavior. A possible solution could be the stratification of the multinomial logistic regression model according to the country of origin.
3. Minor issue
1. Abstract and text: AOR and CI should be defined prior to their usage.
2. Table 1: There’s a typo in the title of the second row: “Respondents with Type 1 DM Glycemic control levels N (%)” should be “Respondents with Type 2 DM Glycemic control levels N (%)”.
3. Table 1: The author should check for the results reported in Table 1. In many cases, percentages do not summing up to 100%.
4. Conclusion
The topic of this paper could be of interest in the diabetes treatment area. However, there are several points that have to be clarified and/or reviewed. In conclusion, I think that this paper is publishable but major revision is needed.
Author Response
1. Summary
In this work, the authors report the results of a multi-national online survey investigating continuous glucose monitoring sensors, smartphone applications usage, and therapy behavior impact on patients affected by type 1 and type 2 diabetes. This study is motivating and offers many ideas to further investigate the topic of interest. Moreover, it is both curious and appealing how the authors used the social media to collect the data. However, I found some weak points that need attention.
Response: We thank the reviewer for their encouraging feedback and insightful reviews which were very helpful to improve this manuscript.
2. Major Comments
1. Line 146: The author should verify and prove that, in Facebook, just clearing search histories avoid the impact of previous searches on future ones. In fact, my guess is that the “Clear searches” feature of Facebook is just a visualization tool to “clear” the search field from listing previous searches. Maybe, the best solution would have been creating two brand new accounts instead of using already existing ones.
In fact one of the accounts we used to do the search were brand new. One of them was existing account. We erased the search history before searching. Erasing the search actually is not only for visualization. Erasing previous search helps to reduce the bias that may impact the search output. We have observed the difference in search outputs before and after erasing our search history.
Response: Users erase their search history mainly to avoid advertisements that are targeted based on their search history and have a better control of their privacy https://www.facebook.com/zuck/posts/10104899855107881. However, it is not clear whether Facebook permanently erases the data from its repository.
2. Line 191: It is not clear how the target variable of the multinomial logistic regression has been defined.
Based on yours and the previous reviewer’s feedback, we updated to elaborated our target variable. Here is a copy of our response to the previous reviewer who has similar concern.
Response: We followed a standard definition to classify glycemic control into three categories. Most of our respondents provided HbA1c values. We also received self-reported capillary glucose levels which the respondents tested themselves at home. For these values, we asked our respondents the timing of their test (pre-prandial or post-prandial). These data were used to categorize the glycemic control levels. Based on the ADA definition, Self-reported HbA1c-levels of < 7.0% or < 53 mmol/mol, or a pre-prandial capillary plasma glucose levels between 80-130 mg/dl or 4.4-4.7 mmol/l, or post-prandial capillary glucose levels<180 mg/dl or <10.0 l="" were="" considered="" as="" good="" glycemic="" control="" levels.="" self-reported="" hba1c-levels="" of="">7.0% or > 53 mmol/mol, or a pre-prandial capillary plasma glucose level >130 mg/dl or >4.7 mmol/l, or post-prandial capillary glucose levels >180 mg/dl or >10.0 mmol/l were classified as hyperglycemia. HbA1c-levels reported as ≤70mg/dl or ≤3.9mmol/l were categorized as hypoglycemia.
As described above we followed the ADA standard definition to categorize the glycemic levels into the three categories. Respondents have provided us their HbA1 values and their capillary blood glucose values. Respondents were asked to provide HbA1c values and to mention where the test was obtained. In addition, data on capillary blood glucose values and the timing of the test (pre-prandial, post-prandial, etc) were also obtained. We also collected data about how frequent respondents have hyperglycemia or hypoglycaemia.
Measuring several episodes of hyperglycaemia and hypoglycaemia is more helpful to understand the glycemic control of patients with diabetes. Determining coefficient of glycemic variation(CV) is an important measure of glycemic variability. However, this is only possible with CGM. We did not have access to CGM data http://care.diabetesjournals.org/content/diacare/40/8/994.full.pdf. We included this point in the limitation section.
3. Line 203: It would be desirable the authors add the description on how data have been prepared in order to train the multinomial logistic regression model.
Response: Per your suggestion, we further elaborated how we prepared the data for analysis.
4. Limitation section: the author should discuss about the fact that it is not possible to be 100% sure that engaging respondents from Facebook groups and online forums, they are all patients with type 1 and type 2 diabetes. Indeed, it could happen that fake accounts are recruited instead.
Response: Yes, we recognize that point. During the data collection, we manually checked the data one-by-one to detect if there are any conflicting responses and removed them from our analysis.
4. One of the major confounders in this analysis is the nationality of the respondents. Indeed, it is well-known that different countries have different dietary regime, culture, and average behavior. A possible solution could be the stratification of the multinomial logistic regression model according to the country of origin.
Response: Yes, that is good way to deal with the heterogeneity of the data. We also wanted to stratify across countries. However, considering the already used two-way stratification level of our analysis ( type 1 and type 2 diabetes with three glycemic levels each) and adding another stratifier (nationality) will result in several modes. That will expand the scope of our paper.
3. Minor issue
1. Abstract and text: AOR and CI should be defined prior to their usage.
Response: Corrected.
2. Table 1: There’s a typo in the title of the second row: “Respondents with Type 1 DM Glycemic control levels N (%)” should be “Respondents with Type 2 DM Glycemic control levels N (%)”.
Response: Thank you for noticing this typographic error. Corrected.
3. Table 1: The author should check for the results reported in Table 1. In many cases, percentages do not summing up to 100%.
Response: Thank you for noticing that rounding errors. We have now corrected to make sure all percentages add up to 100%.
4. Conclusion
The topic of this paper could be of interest in the diabetes treatment area. However, there are several points that have to be clarified and/or reviewed. In conclusion, I think that this paper is publishable but major revision is needed.
Response: Thank you for your insightful reviews. We have revised the manuscript per yours and your fellow reviewers’ feedback. We hope this version is acceptable.
Reviewer 3 Report
The study of Kebebe et al used a multi-national online survey in order to examine factors that are association with glycemic control. Although this is an interesting concept, I have some major concerns:
1) The title is misleading. The authors tried to examine the factors that are associated with glycemic control and not the role of CGM, applications and self care behavior. These were some of the factors that were evaluated, but not all of them. This is evident also in other areas of the manuscript (for example in the introduction section). Please modify accordingly.
2) The definitions of the exposures are also concerning, which represents a major limitation. For hypoglycemia for example:<70 mg/dl is the definition of hypoglycemia, but the frequency of the hypoglycemia should make those subjects to enter this category. A patient, especially with DM1 may have a hypoglycemic value <70 mg/dl , once every 3 months and this is acceptable. However it is not appropriate if this occurs frequently during a week. Similarly about hyperglycemia. The authors used Hba1c (7% cutoff) for definition of eyglycemia and hyperglycemia, which I find appropriate. However they used fasting glucose levels of 80-130 mg/dl and <180 mg/dl for good glycemic control. Fasting levels or postprandial levels can differ through. The authors need therefore to state the frequency of the above events (hopefully they have this information).
3) Another finding is that 95.4% of patients with DM1 are taking glucose lowering medications. Does this mean that the remaining 4.6% were not receiving any type of DM medications? These patients, if they have DM1should develop DKA, without insulin. Or maybe they are on early stage (“ DM honeymoon”). I think that these subjects should not be included in the analysis as they may not have DM1.
4) Race was not included in the models and this should be added as a limitation.
Other minor comments
1) Table 5: Please provide AOR for the type 2 diabetes (for those with poly technique diploma, hyperglycemia vs good glycemic control)
Consider decreasing the length of the introduction section as this is very lengthy,
Author Response
The study of Kebebe et al used a multi-national online survey in order to examine factors that are association with glycemic control. Although this is an interesting concept, I have some major concerns:
1) The title is misleading. The authors tried to examine the factors that are associated with glycemic control and not the role of CGM, applications and self care behavior. These were some of the factors that were evaluated, but not all of them. This is evident also in other areas of the manuscript (for example in the introduction section). Please modify accordingly.
Response: We acknowledge your concern. Our interest was to examine the role of CGM and self-care behaviour and diabetes apps in glycemic control by accounting other confounders. We are primarily interested to highlight the role of these factors. Although our interest is mainly on the three main factors, additional factors that may confound our results are accounted in our analysis. We believe the title better describes the research questions.
2) The definitions of the exposures are also concerning, which represents a major limitation. For hypoglycemia for example:<70 mg/dl is the definition of hypoglycemia, but the frequency of the hypoglycemia should make those subjects to enter this category. A patient, especially with DM1 may have a hypoglycemic value <70 mg/dl , once every 3 months and this is acceptable. However it is not appropriate if this occurs frequently during a week. Similarly about hyperglycemia. The authors used Hba1c (7% cutoff) for definition of eyglycemia and hyperglycemia, which I find appropriate. However they used fasting glucose levels of 80-130 mg/dl and <180 mg/dl for good glycemic control. Fasting levels or postprandial levels can differ through. The authors need therefore to state the frequency of the above events (hopefully they have this information).
Response: This comment was raised by all reviewers. We clarified the definition of the outcome variable to further elaborate it. We followed a standard definition to classify glycemic control into three categories. Most of our respondents provided HbA1c values. We also received self-reported capillary glucose levels which the respondents tested themselves at home. For these values, we asked our respondents the timing of their test (pre-prandial or post-prandial). These data were used to categorize the glycemic control levels. Based on the ADA definition, Self-reported HbA1c-levels of < 7.0% or < 53 mmol/mol, or a pre-prandial capillary plasma glucose levels between 80-130 mg/dl or 4.4-4.7 mmol/l, or post-prandial capillary glucose levels<180 mg/dl or <10.0 l="" were="" considered="" as="" good="" glycemic="" control="" levels.="" self-reported="" hba1c-levels="" of="">7.0% or > 53 mmol/mol, or a pre-prandial capillary plasma glucose level >130 mg/dl or >4.7 mmol/l, or post-prandial capillary glucose levels >180 mg/dl or >10.0 mmol/l were classified as hyperglycemia. HbA1c-levels reported as ≤70mg/dl or ≤3.9mmol/l were categorized as hypoglycemia.
As described above we followed the ADA standard definition to categorize the glycemic levels into the three categories. Respondents have provided us their HbA1 values and their capillary blood glucose values. Respondents were asked to provide HbA1c values and to mention where the test was obtained. In addition, data on capillary blood glucose values and the timing of the test (pre-prandial, post-prandial, etc) were also obtained. We also collected data about how frequent respondents have hyperglycemia or hypoglycaemia.
Measuring several episodes of hyperglycaemia and hypoglycaemia is more helpful to understand the glycemic control of patients with diabetes. Determining coefficient of glycemic variation(CV) is an important measure of glycemic variability. However, this is only possible with CGM. We did not have access to CGM data http://care.diabetesjournals.org/content/diacare/40/8/994.full.pdf. We included this point in the limitation section.
3) Another finding is that 95.4% of patients with DM1 are taking glucose lowering medications. Does this mean that the remaining 4.6% were not receiving any type of DM medications? These patients, if they have DM1should develop DKA, without insulin. Or maybe they are on early stage (“ DM honeymoon”). I think that these subjects should not be included in the analysis as they may not have DM1.
We agree with your point that patients with type 1 diabetes will develop DKA unless they take insulin. Because the response is based in self-response, about 4.6% of respondents with type 1 diabetes did not report taking treatment or data about their treatment history were not available (missing). To make table 1 similar to respondents with type 2 diabetes, we used Yes/No categories for both types of diabetes. We added this variable to the analysis. It did not turn out to be significant. So, the variable doesn’t make any difference.
4) Race was not included in the models and this should be added as a limitation.
Response: Thanks, we already added this variable as one of the unobserved variable in our limitation section. “In addition, participants of the study came from multiple countries. The impact of a broad range of factors, such as differences in healthcare systems and social, cultural, and racial differences were not investigated in the study. These factors potentially limit the applicability of the results to a specific context. However, patients with diabetes who are subscribers of social media are also an important population presenting an interesting paradigm of self-management and social support. Therefore, considering the ever increasing presence and interest of patients with diabetes on social media for seeking diabetes-related information and support, this particular community of diabetes requires attention. In this regard, our research attempted to investigate glycemic control on this particular population. Moreover, the responses of all questions including biochemical parameters are based on self-reported responses. This may also have an impact on our results.”
Other minor comments
1) Table 5: Please provide AOR for the type 2 diabetes (for those with poly technique diploma, hyperglycemia vs good glycemic control)
Response: Thank you for noticing this typographic error. The AOR value is now added.
Consider decreasing the length of the introduction section as this is very lengthy.
Response: The introduction section is now 1 and half page.
Round 2
Reviewer 1 Report
much improved on revision
Author Response
Thanks.
Reviewer 2 Report
The authors fully replied to all the question I asked in a satisfactory way and the manuscript clearly improved its quality. However, I think that one major issue is still present. In fact, my opinion is that stratifying the analysis according to patient nationality is something necessary and required for this work.
For this reason, my conclusion is that this paper is publishable as soon as this point will be satisfied.
Author Response
The authors fully replied to all the question I asked in a satisfactory way and the manuscript clearly improved its quality. However, I think that one major issue is still present. In fact, my opinion is that stratifying the analysis according to patient nationality is something necessary and required for this work.
For this reason, my conclusion is that this paper is publishable as soon as this point will be satisfied.
Response: Yes, that is right. Stratifying the analysis according to nationality will provide mechanisms to handle the heterogeneity due to the unobserved variation that might have been introduced as a result of health system, social and cultural differences. The data comes from more than 50 countries and stratifying the data for this variable was not feasible. However, we are currently working on a further paper by analysing the segment of the data for countries with bigger proportion of responses (USA, UK and Germany, Austria).
We have further added your point on our limitation.
Reviewer 3 Report
The authors tried to revise the manuscript. Although I find the topic very interesting I am concerned about 2 major issues:
1) The definitions of exposures (hypoglycemia, hyperglycemia and normoglycemia ) is problematic. The authors used the ADA definitions , but they did not recognize that a patient can have an Hba1c<7%, glucose values occasionally in the 50s (perhaps one day that he skipped a meal) and also occasionally glucose values in the 300s (when they forget to take insulin, or when a pt had a high crab meal).
I need to emphasize that this concern was raised by all the reviewers.
2) I think that the title is misleading.
Overall I would like to see what other reviewers think, but I am overall concerned about the definitions of the exposures as explained above.
Author Response
The authors tried to revise the manuscript. Although I find the topic very interesting I am concerned about 2 major issues:
1) The definitions of exposures (hypoglycemia, hyperglycemia and normoglycemia ) is problematic. The authors used the ADA definitions , but they did not recognize that a patient can have an Hba1c<7%, glucose values occasionally in the 50s (perhaps one day that he skipped a meal) and also occasionally glucose values in the 300s (when they forget to take insulin, or when a pt had a high crab meal).
Response: Thanks for inquiring further clarification on our outcome definition. Sorry for the confusions. Perhaps, the way we framed our definition was not clear. We have now elaborated our definition. We used cross-sectional data ( a one-time measure of HbA1c and capillary blood glucose level). As you mentioned and also described in the Beck and colleagues 2017 report, HbA1c<7% or="">7% may have a good, hypo- or hyperglycemia. We recognize the variation of glycemic levels that diabetic patients may have even within a single day. HbA1c reflects blood glucose concentration over 3-4 months. However, HbA1c alone might not be a good measure of glycemic levels because of the wide range of glycemic variability corresponding to a given HbA1c level. HbA1 reflects the average value of glucose concentrations. A patient with a given HbA1c may have good, fair and poor glucose levels in one day due to the glycemic variabilities associated with meal time, medication and physical activity. CGM technology is the best way to determine glycemic control and whether HbA1c is over or underestimating glycemic control of patients, ideally with measuring CGM values at least for 14 days. Then averaging the CGM obtained glucose values. But, that is not possible for cross-sectional studies. CGM obtained mean glucose values can then be changed to HbA1c values with a conversion formula (https://www.jaeb.org/gmi/) and compared with a laboratory obtained HbA1c values. The 2017 Beck and colleagues paper on "the fallacy of using averages in using HbA1c" recommended using actual glucose concentration (ideally mean glucose levels from CGM values) combined with HbA1c if patient's level of glycemic control and treatment decisions are to be determined. We don't have access to CGM data. However, we combined both actual glucose values and HbA1c levels for our glycemic control classifications, rather than relying solely on HbA1c or capillary glucose levels.
We included the following point in the limitations section.
"In addition, due to the cross-sectional nature of the study, the results should be carefully interpreted because HbA1c and one-time capillary glucose levels may not be adequate to understand the hyperglycaemic and hypoglycaemic episodes of patients with diabetes. Measuring several glucose levels per day is more helpful to understand the dynamics of glycemic control. Recent studies demonstrated determining the coefficient of glycemic variation (CV) is an important metric to assess glycemic variability than using HbA1c. Ideally, this is only possible with access to CGM."
I need to emphasize that this concern was raised by all the reviewers.
2) I think that the title is misleading.
Response: Per your request, we have now revised the title to “Determinants of Hyperglycemia and Hypoglycemia in Type 1 and Type 2 Diabetes: Results of a Multi-national Online Survey”